# Treatment-seeking behaviour among people with opioid use disorder in the high-income countries: A systematic review and meta-analysis

**Natasha Hall** [1]*, **Long Le**[1], **Ishani Majmudar**[1], **Maree Teesson**[2], **Cathy Mihalopoulos**[1]

**1** School of Health and Social Development, Deakin University, Burwood, Australia, **2** Director Matilda Centre for Research in Mental Health and Substance Use, Sydney University, Sydney, Australia

* Natasha.hall@deakin.edu.au

**Data Availability Statement:** The paper contains the minimal data set, meaning the findings from this study can be replicated in their entirety.

## Abstract

### Objectives

To determine treatment seeking behaviour in those with opioid use disorder (OUD) in the high-income countries.

### Methods

Five databases were searched in November 2019 for quantitative studies that reported OUD treatment seeking behaviour. Data analysis involved determining an overall pooled proportion estimate of treatment seeking behaviour for the two base groups, lifetime treatment and past 12-month or less treatment using the IVhet effect model. Subgroup analysis included heroin OUD, prescription OUD and general OUD. The sensitivity analysis included removal of outliers, separating adults and adolescents and the metaXL sensitivity analysis (studies are excluded if outside the pooled proportion confidence interval of the base case). Systematic review Prospero database registration number [CRD42020159531].

### Results

There were 13 quantitative studies included in the systematic review, with all studies being from the United States of America (USA). IVhet models showed that 40% (95% CI: 23%, 58%) and 21% (95% CI: 16%, 26%) sought treatment in their lifetime and past 12 months respectively. Sub-group analysis found that lifetime treatment seeking for prescription OUD, 29% (95% CI: 27%, 31%), was less than for heroin plus combined OUD, 54% (95% CI: 26%, 82%). Most of the pooled results had high heterogeneity statistics except for results of lifetime treatment seeking for prescription OUD and past 12-month treatment seeking for prescription OUD.

### Conclusion

All included studies in this meta-analysis were from the USA and indicate modest levels of treatment seeking for those with OUD. In particular, this review found that in the USA one in

**Funding:** The author(s) received no specific funding for this work.

**Competing interests:** The authors have declared that no competing interests exist.

five people with OUD sought OUD treatment in the previous 12 months and two in five people with OUD sought OUD treatment in their lifetime. Further research is urgently required to explore the barriers and facilitators that can improve this low treatment seeking in those with OUD.

## Introduction

Opioid use disorder (OUD) worldwide is increasing. From 1990 to 2016 the number of opioid harm related cases worldwide increased by almost 50% [1]. OUD has reached epidemic levels in high income regions such as the United States of America (USA) (1.17%), or Australasia (0.41%) [1]. Compared to the general population, OUDs have been associated with increased risk of premature death [2], increased risk of hepatitis C and human immunodeficiency virus (HIV) [3, 4], increased risk of mental health comorbidities and [3, 5] crime [6], and reduced quality of life [7–9].

Evidence supports opioid agonist treatment (OAT) in terms of reduced illicit opioid use as well as mortality and crime [10, 11]; however, the treatment-seeking behaviour of those with OUD is low. In the USA, less than one in three people with prescription or general OUD sought OUD treatment in the previous year [12–19]. However, it is noteworthy that 81% of those with heroin dependence and 69% of those with heroin abuse sought treatment in the lifetime [20]. An Australian treatment history study of heroin dependence, found that 88% had previously sought treatment in their lifetime [21].

To our knowledge, there is no systematic review and/or meta-analysis that evaluates the proportion of people who are seeking treatment for OUD. Furthermore, the treatment seeking behaviour in those with differing types of OUDs (prescription versus heroin) has not been evaluated. The World Health Organisation (WHO) appraised the current data and found that the treatment gap for alcohol use disorder (AUD) was 78.1%, however no unmet treatment need was provided for OUDs [22]. Therefore, the aim of this review is to determine the proportion of people in high income countries who seek treatment during their lifetime and the last 12-months.

## Method

The systematic review follows Preferred Reporting Items for Systematic Reviews and Meta-Analyses (PRISMA) statement [23]. The protocol for this review was registered on the Prospero database [CRD42020159531].

### Data source

Databases were searched using keywords and Medical Subject Headings (MeSH) on the 28th November 2019. (Appendix A for search terms in S1 File) There was no restriction placed on the publication year. Electronic database searches were conducted in EMBASE and through the EBSCOhost platform for MEDLINE, CINAHL, PsycINFO and SocINDEX databases. Terms relating to OUD treatments (including methadone and buprenorphine), opioid use disorder and treatment-seeking behaviour were included. There was no significant difference in the search strategies in all the databases. Strategies for specific databases only differed on the subject headings used for indexing. Subject headings were searched and uniquely utilised for each database. Grey literature was searched via google scholar using the search term "treatment-seeking behaviour in opioid use disorder" and the reference list of included articles.

Grey literature searching led to the inclusion of 18 further studies for primary screening, of which three articles were included in the systematic review [13, 15, 16].

## Study selection

Studies included in the systematic review were published in English, reported the proportion of treatment-seeking behaviour of those with OUD and described treatment-seeking as seeking any type of treatment for OUD. No limit on publication year was specified. Studies were included where participants had a clinical diagnosis of OUD (using Diagnostic and Statistical Manual of Mental Disorders (DSM) or International Classification of Diseases (ICD) criteria) and where OUD (prescription OUD for those using prescription opioids, heroin OUD for those using heroin and general OUD for unspecified or mixed opioid use) was the primary diagnosis. Qualitative studies, protocol papers, expert reviews or conference abstracts were excluded. Studies where OUD was self-reported by participants were excluded from the review. Studies from non-developed low-income countries as per the United Nations were excluded from the review [24]. Studies from non-developed countries were not included in this review because the treatment seeking for opioid agonist treatment (OAT) is much lower compared with developed countries. Non-developed low-income regions such as Eastern Europe, Latin America and Africa have OAT usage rates of 1%, whereas developed high-income regions such as Australia and North America have OAT usage levels of 23% and 13% respectively [25], and therefore to avoid skewed results only developed countries were included in this review. Further research is required in treatment seeking behaviour in developed countries.

## Data extraction

After removing all duplicates—titles, abstracts and full text were screened by two independent reviewers (NH & IM) and discrepancies were resolved by a third reviewer (LL). Data was extracted from the final included studies and entered into a standardized table. The data extraction sheet included the following categories to describe and compare the studies: author, year, country, sample description, study design, sample size, opioid type, OUD diagnosis tool, participant mean age and gender, type of treatment sought, treatment time frame and percentage or proportion that sought treatment. Independence of all studies was required and therefore one study was not included in the meta-analysis due to the same sample being used in another study [13, 15]. Where the studies reported help-seeking behaviour for both opioid dependence and opioid abuse, opioid dependence was reported. Where studies reported help-seeking behaviour for OUD in differing mental health status groups, no mental health status was chosen. Where studies reported both heroin OUD and prescription OUD, both results were reported in the main analyses and subgroup analyses if appropriate.

## Data analysis

Data analysis involved determining an overall pooled proportion estimate of treatment seeking behaviour. In the primary analysis, the lifetime treatment and past 12 months treatment was conducted. Sub-group analysis and sensitivity analysis were reported. Sub-group analysis involved separating prescription OUD and general OUD (heroin OUD and combined OUD). These distinct groups were decided because treatment outcomes with prescription OUD were found to differ from heroin OUD and combined OUD in terms of more negative urine samples and longer periods of opioid abstinence for prescription OUD compared with heroin and combined OUD [26]. Furthermore, the treatment seeking behaviour and perceived need for treatment was higher for those with heroin OUD (including combined OUD) than for those with prescription OUD [17, 18]. Examination of pooled effect sizes allowed for identification

of outliers. Outliers were defined as studies where the individual effect size and associated 95% CI were outside the 95% CI of the pooled studies (on both CI sides) [27]. The random effects (RE) model and Inverse variance heterogeneity (IVhet) model were chosen and where differences arose, results from both models were presented. If the two models were the same, the IVhet model only was presented while results from the RE model were presented in the Appendices. The IVhet model was chosen because it is an improvement over the RE model and is better able to handle heterogeneity [28]. The RE model provides an underestimation of statistical error and produces overconfident results, which the IVhet model (an estimator model using the fixed effects model assumption that has a quasi-likelihood based variance structure) has been shown to resolve [28].

All data analysis was conducted on Excel using the metaXL as an add-on software package [29]. Statistical heterogeneity between the studies was evaluated using the $I^2$ statistic and Cochran's Q test. As per the Cochrane Collaboration recommendation, heterogeneity was considered important if the $I^2$ statistic was greater than 40% and/or the Q statistic was significant at a P value of 0.01 [30]. Publication bias was assessed using the Doi plot asymmetry index (LFK index). An LFK index of ±1 means no asymmetry, between ±1 and ±2 means minor asymmetry and > ±2 means major asymmetry [29].

### Quality assessment or risk of bias assessment

The studies were assessed for methodological rigour using the Joanna Briggs Institute (JBI) quality assessment tool for cross-sectional studies [31]. The JBI quality assessment tools were used to assess the internal validity and the risk of bias in the studies. The research team determined that good quality studies needed to score 70% or more (score of six or higher out of eight), moderate studies needed to score 50% to 70% (score of four or five out of eight) and poor quality studies scored less than 50% (score of three or less out of eight). Quality assessment was performed on all included studies independently by two reviewers (NH and LL). Any conflicts that arose between the reviewers was resolved by discussion among team members.

## Results

### Study characteristics

There were 13 studies published from 2008 onwards that met the inclusion criteria. See Fig 1 for the PRISMA flow diagram.

Two studies reported results from the same sample population leaving 12 studies included in the meta-analysis. Saha et al. [15] reviewed help seeking in total population whereas Kerridge et al. [13] determined help seeking proportion in men versus woman. Saha et al. [15] was the study included in the overall pooled proportion estimate. All 12 studies were conducted in the USA. Most studies [10/12] used nationally representative cross-sectional surveys in the US including the National Survey on Drug Use and Health (NSDUH) and the National Epidemiologic Survey on Alcohol and Related Conditions (NESARC). Participant recruitment of the remaining two studies [32, 33] occurred via multi-site and single site health care facilities. Eleven of the studies reviewed OUD in adults and two of the studies included only adolescents in the sample [12, 33]. The key characteristics of the included studies can be found in Appendix B in S1 File.

### Quality assessment of the included studies

Most of the included studies [11/12], were of good quality according to the JBI checklist. The remaining study was of moderate quality [33]. The quality assessment score for each included study can be seen in Appendix B in S1 File.

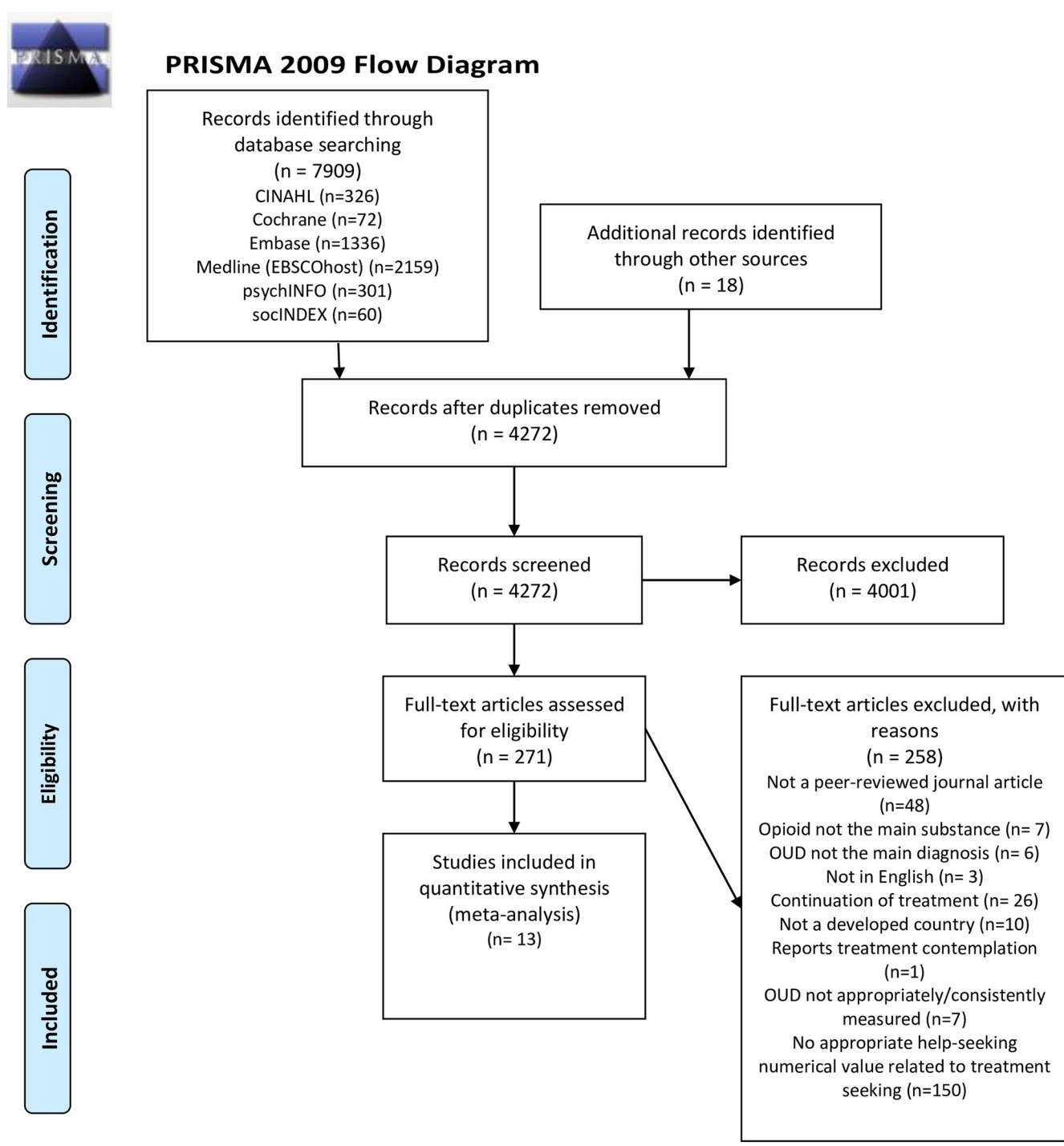

From: Moher D, Liberati A, Tetzlaff J, Altman DG, The PRISMA Group (2009). *Preferred Reporting Items for Systematic Reviews and Meta-Analyses: The PRISMA Statement.* PLoS Med 6(7): e1000097. doi:10.1371/journal.pmed1000097

**For more information, visit www.prisma-statement.org.**

**Fig 1. PRISMA flow diagram.**

## Meta-analysis base case results

The included studies investigated OUD help-seeking behaviour from lifetime treatment and 12-month or less treatment. These time frame differences contribute to increased heterogeneity between the studies and therefore the base cases examined for this meta-analysis will be lifetime treatment seeking and 12-month or less treatment seeking.

### Lifetime OUD treatment seeking

The RE model (Appendix C in S1 File) and IVhet models produced similar results and therefore the IVhet model will be used. Fig 2 shows that 40% (95% CI: 23%, 58%) of people with OUD sought help for OUD treatment in their lifetime. The $I^2$ statistic for heterogeneity is 98%, which suggests a considerable amount of difference between the studies. The five studies included in the lifetime treatment seeking base group were cross-sectional studies from online recruitment samples [33] or nationally representative surveys [15, 18, 20, 34]. Two of the studies determined help-seeking behaviour for prescription OUD [15, 34], one study reviewed OUD help-seeking behaviour for all opioids [18] and the final study reviewed treatment seeking from both heroin and prescription opioids [20]. One of the included studies was undertaken in adolescents only [33] and in the remainder of the studies the population was adults. The LFK index, which reports publication bias, reported no asymmetry with LFK = -0.14. (see Appendix D in S1 File).

### 12-month or less OUD treatment seeking

The RE model (Appendix E in S1 File) and IVhet models were comparable and therefore only the IVhet model was used. Fig 3 shows that of those with OUD, 21% (95% CI: 16%, 26%) sought treatment in the past 12 months. The 95% CI interval is small, there are several outliers and the $I^2$ statistic is high [97%]. Eight of the nine studies included used a nationally representative survey and examined help-seeking over the past 12 months, the remaining study was a multi-site study and reviewed help-seeking behaviour in the past 6 months [32]. The type of OUD that was present in the included studies were prescription OUD [12, 14, 15], combined OUD [16–19, 35] and one study reported both prescription OUD and heroin OUD [32]. One of the included studies was undertaken in adolescents only [12] and the remaining eight studies were all done in adults. The LFK index for model 2 reported major asymmetry with LFK = -2.01. (see Appendix F in S1 File).

### Meta-analysis subgroup results

Subgroup analysis, where the different types of OUD were grouped together, was undertaken. These subgroups included prescription OUD and heroin plus combined OUD. These two subgroup analyses were done within the two base groups.

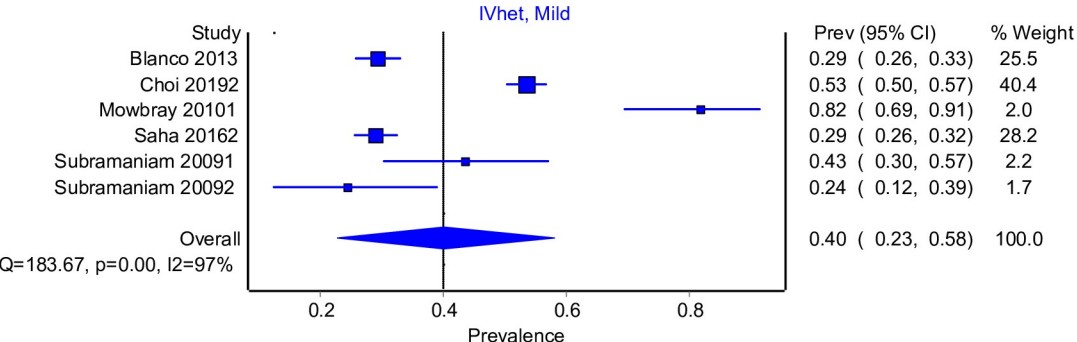

**Fig 2. Lifetime treatment seeking for OUD IVhet model.**

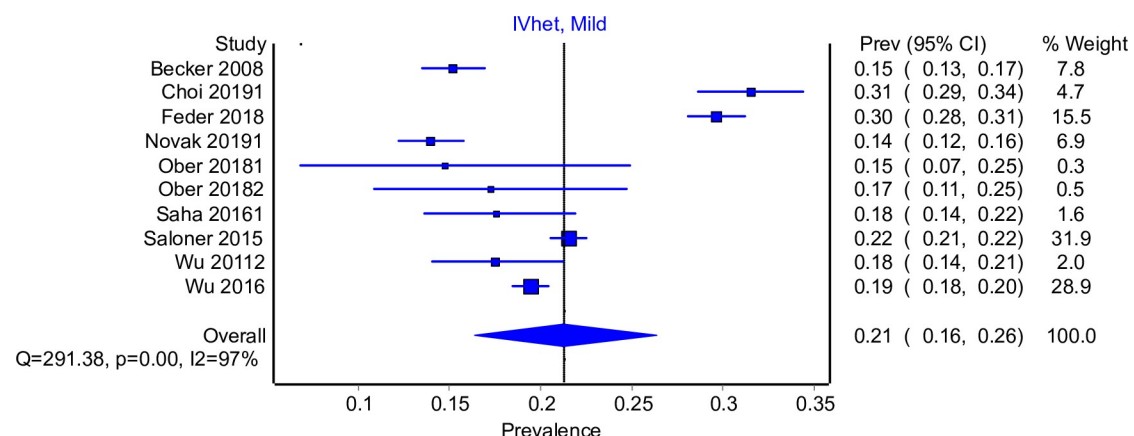

**Fig 3. 12-month or less treatment seeking for OUD IVhet model.**

## Lifetime treatment seeking

Lifetime treatment-seeking proportion of those with OUD with primary heroin use plus combined OUD included 3 studies [18, 20, 33] (Appendix G in S1 File). The treatment-seeking proportion was 54% [95% CI: 26%, 82%], with a high $I^2$ statistic (90%). Two of the studies used national surveys and included adults [18, 20] and the third was a single treatment site including only adolescents [33]. The treatment-seeking proportion of those with lifetime treatment history for prescription OUD was 29% (95% CI: 27%, 31%) (see Appendix H in S1 File), with no heterogeneity ($I^2$ = 0%).

## 12-month or less treatment seeking

Six studies [16–19, 32, 35] examined 12-month treatment seeking with heroin plus combined OUD. The treatment seeking for those with heroin plus combined OUD over the past 12 months was 22% (95% CI: 17%, 28%, $I^2$ = 98%). (Appendix I in S1 File) One study examined heroin OUD only [32] and five studies examined combined OUD [16–19, 35]. Five studies used the NSDUH survey and the remaining study used a multi-site sample [32]. All six studies included adults only.

The treatment seeking percentage for those with prescription OUD over the past 12-months was 16% (95% CI: 15%, 17%) (see Appendix J in S1 File), with no heterogeneity ($I^2$ = 0%) of 0%. The past 12-month treatment seeking proportion for general OUD was 22% (95% CI: 16%, 18%) (see Appendix K in S1 File). The heterogeneity was high between the five studies, with the $I^2$ statistic being 98%. The studies were all undertaken in adults and used nationally representative surveys, however because all opioids were included the varying types of OUD may be contributing to heterogeneity.

## Meta-analysis sensitivity analysis results

Due to high heterogeneity, three sensitivity analyses were undertaken. These were to remove the outliers, separate adults from adolescents and lastly the metaXL sensitivity analysis. The outliers were removed in both the base case models; lifetime treatment seeking for OUD and 12-month treatment seeking for OUD.

One outlier was outside the 95% CI range and was removed from the lifetime treatment base case (see Appendix L in S1 File) [20]. The help-seeking percentage, 39%, (95% CI: 23%, 56%) did not change with removal of the outlier, due to the small weight (2%) the outlier had

on the overall pooled proportion. $I^2$ statistic was 98%, indicating significant heterogeneity. The outlier study included individuals with heroin OUD, used a nationally representative sample and had a small sample size (n = 150). The included studies were related to prescription OUD [15, 33, 34], and general OUD [18]. Three of the included studies were nationally representative samples [15, 18, 34] and the remaining study was a single site sample [33].

Four outlier studies were removed from the 12-month treatment seeking base case [14, 18, 19, 35] and in one study the prescription OUD treatment seeking was an outlier, but not the heroin OUD treatment seeking proportion [32]. (Appendix M in S1 File) Removal of the outliers reduced the treatment seeking percentage from 21% to 20% and narrowed the confidence interval. The $I^2$ statistic reduced from 97% to 69% with removal of the outliers [14, 18, 19, 32, 35]. The outlier studies were related to prescription OUD [14, 32] and general OUD [14, 19, 35] and all used nationally representative samples. The included studies were reviewing treatment seeking in those with prescription OUD [12, 15], combined OUD [16, 17] and heroin OUD [32].

The second sensitivity analysis was to separate adults and adolescents within the two base case models; lifetime treatment seeking for OUD and 12-month treatment seeking for OUD.

## Lifetime OUD treatment seeking

Four studies were included when the sensitivity analysis was adults only lifetime treatment seeking (see Appendix N in S1 File). The pooled percentage of treatment seeking was 40% (95% CI: 22%, 59%), with significant heterogeneity ($I^2$ = 98%). The included studies were identifying treatment seeking in prescription OUD [15, 34], heroin OUD [20] and general OUD [18]. Four of the five included studies used nationally representative samples and the remaining study used an online opportunistic sample [16]. One study examined lifetime OUD treatment seeking in adolescents [33] with treatment seeking proportions for both heroin OUD and prescription OUD. The treatment seeking was 35% (95% CI: 17%, 54%, $I^2$ = 72%). (Appendix O in S1 File) The study used a single-site sample.

## 12-month OUD treatment seeking

Eight studies were included in this sensitivity analysis for adults only (see Appendix P in S1 File). One study [12] was not included in this model when compared with the 12-month treatment seeking base case. Hence the pooled proportion is the same as the base case, 21% (95% CI: 16%, 27%, $I^2$ = 97%). The included studies were identifying treatment seeking in prescription OUD [14, 15, 32], combined OUD [16–19, 35] and one study reported for both heroin OUD and prescription OUD [32]. Seven of the studies used nationally representative surveys and one study used a multi-site survey [32].

Only one study reviewed 12-month treatment seeking in adolescents [12], which means a sensitivity analysis could not be completed.

Lastly, the final sensitivity analysis was using the metaXL sensitivity analysis for the two base cases; lifetime treatment seeking for OUD and 12-month treatment seeking for OUD. Results will only be reported if the pooled proportion value for the excluded study is outside the pooled proportion confidence interval of the base case.

Table 1 show the metaXL sensitivity analysis where the pooled proportion is given when the specified study is removed in turn. This sensitivity analysis found no outlier cases. The pooled proportion lifetime treatment seeking was 31% to 45%.

Table 2 show the metaXL sensitivity analysis where the pooled proportion is given when the specified study is removed in turn. This sensitivity analysis reported no outlier cases. The pooled proportion 12-month treatment seeking was 20% to 22%.

**Table 1. Lifetime OUD treatment seeking IVhet base model metaXL sensitivity.**

| Removed each study in turn | Pooled prevalence | LCI 95% | HCI 95% | Cochran Q | P | I² | I² LCI 95% | I² HCI 95% |
|---|---|---|---|---|---|---|---|---|
| Blanco 2013 | 0.438 | 0.201 | 0.683 | 140.864 | 0.000 | 97.160 | 95.348 | 98.267 |
| Choi 2019 | 0.312 | 0.163 | 0.472 | 62.104 | 0.000 | 93.559 | 87.877 | 96.578 |
| Mowbray 2010 | 0.391 | 0.231 | 0.557 | 145.355 | 0.000 | 97.248 | 95.513 | 98.312 |
| Saha 2016 | 0.445 | 0.208 | 0.689 | 131.828 | 0.000 | 96.966 | 94.977 | 98.167 |
| Subramaniam 20091 | 0.399 | 0.218 | 0.587 | 183.388 | 0.000 | 97.819 | 96.565 | 98.615 |
| Subramaniam 20092 | 0.402 | 0.225 | 0.587 | 179.294 | 0.000 | 97.769 | 96.475 | 98.588 |

## Discussion

Our systematic review and meta-analysis found that one in five people (20%) or one in two people (50%) with OUD sought help during the previous 12-months and during their lifetime in the USA, respectively. It is noteworthy that it is the first review including a meta-analysis to quantify the proportion of people with OUD seeking treatment. A study conducted by the World Health Organisation (WHO) found that the 12-month treatment seeking proportion for mood disorders and substance use disorders in the USA was 35% and 10% respectively [36]. The treatment seeking proportion for OUD in our study is therefore slightly less than the treatment seeking for mood disorders and slightly higher than the treatment seeking for substance use disorders [36]. The lower treatment seeking levels in those with substance use disorders could be due to lack of established effective and available treatments for other substance use disorders such as cocaine and methamphetamine compared to available and effective opioid replacement treatments for OUDs [37]. It is expected that treatment seeking occurs at a higher proportion within a longer timeframe (12 months versus lifetime) and other studies have also found that increased disease duration and severity increase the likelihood of treatment seeking for OUD [15, 21, 32]. Treatment seeking for prescription OUD was the lowest (16% for 12-month treatment and 29% for lifetime treatment) and for heroin plus combined OUD was the highest (54% for lifetime treatment). Reduced treatment seeking in those with prescription OUD when compared with heroin OUD could be explained by the difficulty in diagnosing prescription OUD due to co-existing complex pain management needs and long term opioid use [38]. Secondly, lower levels of treatment seeking in those with prescription OUD may be related to treatment appropriateness. Although treatment seeking for opioid agonist treatment (OAT) has been demonstrated to be effective for prescription OUD [39], the differing characteristics between prescription OUD and heroin OUD [40, 41] (together with OUD treatment being developed primarily for illicit drug users) suggests that treatment

**Table 2. 12-month OUD treatment seeking IVhet base model metaXL sensitivity.**

| Removed each study in turn | Pooled prevalence | LCI 95% | HCI 95% | Cochran Q | P | I² | I² LCI 95% | I² HCI 95% |
|---|---|---|---|---|---|---|---|---|
| Becker 2008 | 0.218 | 0.166 | 0.272 | 247.045 | 0.000 | 96.762 | 95.329 | 97.755 |
| Choi 2019 | 0.208 | 0.161 | 0.257 | 234.927 | 0.000 | 96.595 | 95.060 | 97.653 |
| Feder 2018 | 0.198 | 0.156 | 0.242 | 147.481 | 0.000 | 94.576 | 91.666 | 96.469 |
| Novak 2019 | 0.218 | 0.169 | 0.270 | 233.948 | 0.000 | 96.580 | 95.037 | 97.644 |
| Ober 20181 | 0.213 | 0.164 | 0.264 | 289.910 | 0.000 | 97.241 | 96.091 | 98.052 |
| Ober 20182 | 0.213 | 0.163 | 0.264 | 290.328 | 0.000 | 97.244 | 96.097 | 98.054 |
| Saha 2016 | 0.213 | 0.163 | 0.265 | 288.621 | 0.000 | 97.228 | 96.072 | 98.044 |
| Saloner 2015 | 0.211 | 0.146 | 0.280 | 290.997 | 0.000 | 97.251 | 96.107 | 98.058 |
| Wu 2011 | 0.213 | 0.163 | 0.266 | 287.549 | 0.000 | 97.218 | 96.055 | 98.038 |
| Wu 2016 | 0.22 | 0.155 | 0.289 | 274.016 | 0.000 | 97.080 | 95.838 | 97.952 |

options, and therefore treatment-seeking may be less acceptable or accessible to those with prescription OUD. For example, those with prescription OUD often have associated pain [41, 42]. Opioids are used to treat chronic pain and therefore those with prescription OUD experience fear that seeking OUD treatment will lead to uncontrolled pain [43]. The fear of uncontrolled pain requires attention so that those with dual pain and OUD can be better engaged in OUD treatment. Those with prescription OUD were also more likely to access health services, be prescribed antidepressants, use benzodiazepines, were older and had higher anxiety scores than those with heroin OUD [40–42, 44]. This emphasises that treatment compliance is not the problem and that reduced treatment seeking for prescription OUD is related to another factor. Furthermore, depression as a comorbidity can exacerbate the experience of pain, leading to increased need for pain relief [45, 46]. Stigma may also play a role in reduced treatment seeking for those with prescription OUD. A recent systematic review found that stigma was the third most common reported barrier to OAT [47]. Those with prescription OUD experience addiction and treatment stigma in different ways than those with heroin OUD [43]. An Australian study found that those with prescription OUD thought OAT was for heroin users or for illicit drug users, which creates a separation between themselves (prescription opioid users) and heroin users [48]. Prescription and heroin OUD related stigma may be reduced by improving community understanding of addiction and reducing societal impacts of drug use such as crime (via harm reduction services such as safe injecting rooms) [49].

All studies in this systematic review were from the USA. It is noteworthy that other high-income countries report proportion of seeking treatment in substance use disorders in general, but not specific to opioid use disorders. Australian studies using the National Survey of Mental Health and Wellbeing found that one third of those with a SUD sought professional help in the past 12 months in 1997 [50] and 11% (16–24 year-olds), 32% (25–44 year-olds) and 29% (45–85 year-olds) sought professional help in the past 12 months in 2007 [51]. Therefore this review, with studies only from the USA, found that less people with an OUD sought treatment (one in five people with OUD sought treatment in the past 12 months) than those with SUD in Australia (one in three people with SUD sought treatment in the past 12 months). Treatment seeking within 50 years of SUD onset was examined by Western Europe, New Zealand and Japan. In Western Europe lifetime treatment seeking ranged from two in five (Spain) to four in five (Germany) [36]. In New Zealand and Japan, the lifetime treatment seeking proportion was four in five and three in ten, respectively [36]. The 50-year treatment seeking results for SUD reported in these regions are mostly higher (except for Spain and Japan) than found in our review, which reported that one in two people sought treatment for OUD over their lifetime. Treatment programs, differing health systems, stigma in differing culture as well as stigma surrounding specific drug used all play a role in treatment seeking behaviour. In contrast to the USA, European countries, such as Germany and Portugal, have decriminalised drugs for personal use, have medically supervised injecting rooms and have many OUD treatments (including heroin assisted treatment) covered by public health insurance [52–54]. These steps have been shown to increase treatment uptake for OUD [52, 53]. The treatment of OUD as a health condition by these countries has the potential to further reduce stigma by improving community understanding of addiction [49]. Furthermore, stigma is experienced differently with different drugs. Internalised stigma is heightened in heroin users compared with marijuana [55], alcohol and methamphetamine users [56]. Perceived stigma is heightened towards those who use opioids non-medically compared to alcohol, marijuana, ecstasy, and other stimulant users [56, 57]. More research on OUD specific treatment seeking in countries other than the USA is urgently required to better understand treatment seeking within differing countries with differing health systems.

The heterogeneity in most of the models was high and contributing factors are variability in OUD type, age of participants and methodological differences. Heterogeneity was low ($I^2$ = 0%) in two models, which evaluated prescription OUD during the past 12-months and life-time. Lifetime treatment-seeking in those with OUD with primary heroin use plus combined OUD was the highest, but also had high heterogeneity ($I^2$ = 90%), a wide confidence interval and included only two studies in the meta-analysis, with one study reporting both prescription OUD and heroin OUD. These results imply that treatment seeking is more uniform in those with prescription OUD compared to heroin OUD. In the USA prescription OUD often pre-dates heroin OUD [58, 59], which may allow health practitioners to notice and target prescription OUD in a more uniform way compared to if the client progresses to heroin OUD. Furthermore, one American study found that the prescription OUD group reported higher rates of other prescription drug abuse than the heroin OUD group [12]. The more regular and continuous contact with health professionals to access prescription medications could further explain the more uniform treatment seeking patterns in those with prescription OUD. More research is required to determine the true treatment seeking proportion for those with heroin OUD.

There is more research of treatment-seeking behaviour for OUD in adults. Only one study each examined 12-month and lifetime treatment-seeking behaviour for prescription OUD in adolescents and hence a meta-analysis model was not undertaken. This study indicated that 12-month and lifetime treatment seeking in adolescents for prescription OUD was 18% [12] and 24% [33] respectively, which is less than what is found in adults. These results imply that adolescents are less likely to seek treatment than older adults, which has been previously demonstrated in the literature [17, 60–62]. One reason may be because adolescents may have early disease progression and/or are less severe in the addiction trajectory compared to adults. Furthermore, reduced help seeking may be due to increased stigma and fear experienced in younger cohorts, low health literacy, less disease exposure, or even not knowing that there might be effective treatments available. Regardless, more research in needed in OUD treatment seeking behaviour of adolescents.

Several barriers to treatment seeking for OUD have been identified in the literature. These barriers are likely to contribute to the low treatment seeking rates we have found in the current study. Barriers to treatment seeking include: social stigma [63–73]; treatment cost [70, 74, 75]; perceived lack of flexibility around treatment [67, 74, 76, 77]; a lack of prescribers [64, 75, 78–80]; long waiting lists [61, 64, 67, 73, 75, 81–85]; geographical barriers [86, 87]; psychosocial barriers (feelings of worthlessness and low self-esteem) [64, 84, 88]; chaotic lifestyle [60, 61, 64, 86, 88–90]; cultural barriers [60, 66, 71, 84, 87, 89]; and finally, a lack of support services (psychologists) [70, 72, 80, 84]. Treatment programs need to be developed, which help to reduce these barriers and therefore encourage all those with OUD to seek treatment. The low levels of treatment seeking found in the USA in this systematic review are likely a combination of inadequate access to treatment programs and psychosocial challenges, which differ between countries with different health systems. For example, in the USA, only 13.8 percent of OUD treatment programs accepted Medicare and covered any Food and Drug Administration approved opioid use disorder medication [91]. This is compared with the UK, where the National Health System (NHS) fully covers any medications for OUD approved and recommended by the National Institute for Health and Care Excellence (NICE) [92]. More research is required to determine the most important barriers to OUD treatment from the perspective of both prescription OUD and heroin OUD in other countries.

## Literature gap

The exclusion criteria for this review includes several exclusion categories that may show where the literature gaps are when studying treatment seeking behaviour in other countries (USA and non-developed countries are excluded). There was one study from the Netherlands where opioid use was not the main substance studied and rather SUD more generally was studied [93]. Similarly, there were two studies from the United Kingdom [94, 95] where OUD was not the main diagnosis and SUD more generally was studied. This emphasises the need for more specific OUD related treatment seeking behaviour. Furthermore, there are no reviews looking at treatment seeking in substance use disorders more generally indicating this is an area for further research. OUD diagnosis was not measured using a reliable diagnostic tool, for example the ICD-10 or the DSM-V, in studies from United Kingdom [96, 97], Canada [98] and Australia [99] meaning that a diagnosis is based on self-report or other unvalidated measures (e.g. self-report of illicit drug use in the past month or the presence of injecting marks).

## Literature limitations

All included studies were from the USA, which means the results cannot be generalised to other countries due to the notable differences in health care systems, which may lead to differences in treatment seeking behaviours for those with OUD. In the USA healthcare is a market driven business opportunity whereas in other countries, for example the UK, healthcare is a public good [100]. It would be ideal to include other countries with varying health systems in the review, however presently these studies are not available in the literature. Secondly, all the included studies used cross-sectional surveys (see Appendix A in S1 File) and all but one study [32] assessed treatment seeking through self-reporting, which has the potential to introduce recall bias regarding treatment. Furthermore, although only studies with DSM and ICD OUD diagnoses were included in this review, self-reporting is used to input the data for these diagnostic tools, which may introduce recall bias. Lastly, ten of the 13 studies used the NSDUH and NESARC differing year surveys to provide nationally representative samples, however these surveys do not include homeless individuals. Those with OUD are often homeless and therefore this may underestimate the prevalence of treatment seeking for OUD in this vulnerable population.

## Study limitations

Although this review included studies that were conducted in developed countries, all studies were from the USA. Therefore, results are not transferable to countries other than the USA. More research is required on the treatment seeking levels in other countries. Secondly, the review only included studies published in English and therefore important treatment seeking studies published in other languages have been missed. Thirdly, the treatment seeking timeframe reported varied between studies and included past 6 months, 12-month and lifetime treatment seeking. This bias was able to be reduced by undertaking subgroup analysis of the results. Fourthly, treatment seeking in this review was treatment or counselling related to OUD. However, one study involved only pharmacological treatment for OUD [32] and the second included both OUD treatment and psychiatric treatment, which could not be separated [33]. Mostly, treatment was similar, however these minor differences in treatment items between studies could have introduced some bias. Lastly, although broad terms relating to OUD treatment and pharmacotherapy were used in the search strategy, the only specific treatment types included were methadone and buprenorphine. This means that studies with other specific treatment types, for example behavioural interventions or naltrexone, may have been missed. However, this review is determining the prevalence of treatment seeking to any

treatment rather than individual treatment types and therefore the results may not be heavily impacted.

## Conclusion

The overall pooled proportion of those with a lifetime OUD treatment history was about 40% and those with a 12-month OUD treatment history was around 20%. These results indicate low levels of treatment seeking for those with OUD in the USA. The reasons for these modest levels of treatment-seeking behaviour and barriers to treatment in this USA population require immediate attention and future work should be undertaken in this area. Treatment-seeking behaviour in the USA was found to be lowest in adolescents and adults with prescription OUD and highest in adults with heroin OUD.

## Supporting information

**S1 Checklist. PRISMA 2009 checklist.**
(DOC)

**S1 File. Contains all the supporting tables and figures.**
(DOCX)

## Author Contributions

**Conceptualization:** Natasha Hall, Long Le, Cathy Mihalopoulos.

**Formal analysis:** Natasha Hall.

**Investigation:** Natasha Hall, Ishani Majmudar.

**Methodology:** Natasha Hall, Ishani Majmudar.

**Project administration:** Natasha Hall.

**Resources:** Natasha Hall, Ishani Majmudar.

**Software:** Natasha Hall.

**Supervision:** Long Le, Cathy Mihalopoulos.

**Writing – original draft:** Natasha Hall.

**Writing – review & editing:** Natasha Hall, Long Le, Maree Teesson, Cathy Mihalopoulos.

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
