## [Decision Letter · Decision Letter 0]

15 Feb 2021

PONE-D-20-31370

Treatment-seeking behaviour among people with opioid use disorder in the high-income countries: A systematic review and meta-analysis

PLOS ONE

Dear Dr. Hall,

Thank you for submitting your manuscript to PLOS ONE. After careful consideration, we feel that it has merit but does not fully meet PLOS ONE’s publication criteria as it currently stands. Therefore, we invite you to submit a revised version of the manuscript that addresses the points raised during the review process.

However, this decision does not guarantee final acceptance. There seem to be several important issues that need to be clearly resolved by revision. Particularly, read through the comments of the reviewer #2 and #3 and then respond to them carefully.

We look forward to receiving your revised manuscript.

Kind regards,

Kyoung-Sae Na, M.D.

Academic Editor

PLOS ONE

Journal Requirements:

Reviewers' comments:

Reviewer's Responses to Questions

**Comments to the Author**

1. Is the manuscript technically sound, and do the data support the conclusions?

Reviewer #1: Partly

Reviewer #2: Yes

Reviewer #3: No

2. Has the statistical analysis been performed appropriately and rigorously? 

Reviewer #1: Yes

Reviewer #2: Yes

Reviewer #3: Yes

3. Have the authors made all data underlying the findings in their manuscript fully available?

Reviewer #1: Yes

Reviewer #2: Yes

Reviewer #3: Yes

4. Is the manuscript presented in an intelligible fashion and written in standard English?

Reviewer #1: Yes

Reviewer #2: Yes

Reviewer #3: Yes

5. Review Comments to the Author

Reviewer #1: The current manuscript entitled “Treatment seeking behaviour among people with opioid use disorder in high income countries: A systematic review and meta-analysis” evaluates treatment seeking behavior in individuals with OUD. The protocol for the systematic review and meta-analysis appear sound and maintain rigor (although see point 1 below). The findings are interesting and add to the body of evidence that healthcare providers and programs need to do more to reduce barriers to OUD care and medication for OUD. There are a number of weakness that diminish enthusiasm and need to be addressed should the authors be invited to submit a revision. Most importantly, there is one article that was included but does not appear to meet inclusion criteria (see point 3 below). This may change the results and limits the scope to US populations. There are also some stigmatizing elements of the discussion that are unsupported or based on extremely outdated evidence. Major and minor issues detailed below.

Major

1) The manuscript stated that the search was performed on 11/28/2019 but the PROSPERO protocol was created on 04/20/20. This alone is not necessarily a problem, but registering after starting literature search is not consistent with PRISMA guidelines. The authors should state that the protocol was pre-registered after search was completed and clearly state what aspects of the protocol were completed prior to registration on PROSPERO. It is possible that registration was in response to a prior review at another journal, but the authors should be transparent.

2) Please consider using “medications for Opioid Use Disorder” instead of OST or, to better describe buprenorphine and methadone, use opioid agonist treatment.

3) The authors state that only studies that included DSM-5 or ICD diagnosed opioid use disorder were included and studies that relied on self-reported OUD were excluded. However at least one study (Kimergard et al., 2017) appears to have used online surveys from social media and treatment websites. It is easily assumed that the participants in these surveys self-reported OUD because the surveys were online links. In fact, the authors state that Kimergard used the Severity of Dependence Scale which is 5 items and, while potentially correlated with DSM and ICD OUD diagnosis, does not equate to an OUD diagnosis. Without this study, all the other studies are in the US which would change the scope of the article (i.e., not high income countries but rather just U.S.). This manuscript would appear to call into the full-text exclusion category of “OUD not appropriately/consistently measured”.”

4) “Heroin use disorder” is not a diagnosis; OUD with primary heroin use is more appropriate. The authors also separate findings into prescription OUD, heroin OUD, and general OUD. This may be unnecessarily confusing. It appears the difference between prescription OUD and non-prescription OUD (i.e. heroin and/or general OUD) is most informative with respect to treatment engagement. Also, the heterogeneity in the prescription OUD is substantially less than the other OUD categories. This is potentially useful and should be discussed more that it is in the findings. Why are the rates of treatment higher and more uniform in prescription OUD versus other OUD categories?

5) The study column in Table 1 and 2 states “Excluded Study.” These studies were not excluded though. Please clarify.

6) Discussions about treatment gap for the Europe are inappropriate because there is only one study (only UK and Ireland) and it appears this study should be excluded (see point 2 above). Even if the authors can justify including Kimergard, this single study is in no way representative of Europe.

7) The authors postulate that individuals with higher rates of OUD and primary heroin use are higher than prescription OUD because of the “heavy disruption of heroin use” on daily life. This is not always the case and potentially not related at all to treatment engagement. The authors also state that “heroin users often seek treatment after hitting ‘rock bottom’ and treatment is often initiated after court orders.” This language is incredibly stigmatizing and potentially inaccurate. In fact, the authors cite a paper from 1971 to justify this statement. Higher rates of treatment seeking for individuals with OUD and primary heroin use may be more likely due to stigma of heroin use and challenges diagnosing prescription OUD particularly in the context of long term opioid treatment for chronic pain (see Manhapra et al. (2020). Complex Persistent Opioid Dependence with Long-term Opioids: A Gray Area That Needs Definition, Better Understanding, Treatment Guidance, and Policy Changes. Journal of General Internal Medicine, 1-8.).

8) The authors state that individuals with prescription OUD are more likely to have comorbidities compared to those with OUD and primary heroin use. The type of comorbidities should be discussed. Medical comorbidities in individuals with prescription OUD may be driven by co-occurrence of chronic pain, which is associated with considerable multimorbidity.

9) The authors discuss lower treatment engagement in the current findings compared to general SUD in other developed nations. The authors fail to discuss possible reasons beyond different access to care across countries. This appears to be a great opportunity to talk about potential sigma as a barrier to treatment that may be specific to OUD and review ways to reduce barrier.

Minor

10) There are numerous awkward phrasing and grammatical issues in the text, most notably in the results. For example, in the results “Lifetime treatment-seeking proportion of those with heroin use disorder included 2 studies only and the treatment seeking proportion was 65% with a wide confidence interval as well as high I2 statistic, both indicating heterogeneity between studies.” This sentence contains a lot of information and is very difficult to follow. Suggest the authors structure reporting the findings in the same way and stick to a consistent presentation of findings. It is sometimes confusing which studies are connected to a reported I2 statistic.

11) Unclear why there are numbered commented in the meta-analysis sensitivity results.

12) Meta-analysis result figures in the supplemental materials are cut off on the right side. Please revise to include complete figure.

Reviewer #2: I found this to be a very interesting and well written article. My comments are relatively minor:

- Abstract. Please review the Objective sentence. There appears to be a word missing. Maybe "behaviour"?

Reviewer #3: This manuscript sought to understand the “gap” in treatment-seeking behavior among individuals with opioid use disorder in high-income countries. While a noble effort and important topic, there are significant concerns in the presentation of the information, as well concerns about biases in the data collection that warrant further understanding.

The title is a little misleading in that the review doesn’t cover “high-income countries” as a whole. Despite that being the intent, the data are nearly entirely focused on the United States, and so there is a potential for readers or others to generalize these results to other high-income countries where there are significant differences in healthcare compared to the United States. Given that the end result in the data, the authors may consider removing the single UK study and focusing the paper on the United States alone, which would still be of interest. The original methodology and research question could still be included, but with the decision given the search results, that the instead the data were limited to the United States.

The authors need to provide better context surrounding the complexity of the gap in treatment-seeking for opioid use disorder. The authors note previous research identifying gaps in treatment-seeking for alcohol use disorder, but opioid use disorder involves more complex substance use patterns that make it more difficult to account for treatment-seeking patterns, particularly when comparing 2008, the year of the first study reported in the analysis, to that of recent years.

Most notably, this span of years in the United States is complicated by two issues:

1. In 2008, opioid use predominantly involved prescription opioids, but over the past decade, heroin use drastically increased in the wake of prescription opioid use reductions resulting from prescription-focused interventions.

2. In addition, the provision of buprenorphine was extremely low early on, but has drastically increased, making treatment with buprenorphine not equally accessible across the study time period.

Even today, many treatment centers/programs in the United States, due to our disparate provision of healthcare, do not provide buprenorphine or methadone. According to the 2018 National Survey of Substance Abuse Treatment Facilities, 39.6% provide no pharmacotherapy treatment. Although the authors appear to use broad terms for opioid treatment, the lack of searching for other specific types of treatment like the inclusion of buprenorphine or methadone may have limited the results.

In addition, using only articles in English would also have been a significant factor in limiting the results. Not all high-income countries publish all of their articles in English, and suggesting there is not data from high income countries is different than saying there is not data published in English from high income countries.

It may be helpful to understand the exclusion criteria in the flow chart at a country level. This could help inform readers as to what the current limitations are, or what types of studies are being conducted in other countries, in order to spur more research on this issue in those countries.

The authors note that self-reported studies were excluded from the interview. However, they include data collected from NSDUH, which is self-reported. I believe NSDUH was included because DSM criteria are included in the questions, thus offering a way to “meet” diagnostic criteria, but this is still self-report.

“Where studies reported both heroin OUD and prescription OUD, prescription OUD was reported in the main analyses while heroin OUD was included in the subgroup analyses?” What was the rationale for this decision? This seems like it could have time-related bias.

11 of the 14 studies stem from two surveys NESARC and NSDUH, which suggests the results may be more indicative of these two national survey in the United States rather than “treatment-seeking behavior of high-income countries” and so these results would be limited in the same the way the data from these surveys are limited. The authors may consider limiting their paper to just the United States, where a synthesis of this data would be more meaningful.

6. PLOS authors have the option to publish the peer review history of their article (what does this mean?). If published, this will include your full peer review and any attached files.

Reviewer #1: No

Reviewer #2: No

Reviewer #3: No

---

## [Author Response · Author response to Decision Letter 0]

3 Mar 2021

Please see word document attached earlier titled " PLos one response to reviewers"

---

## [Decision Letter · Decision Letter 1]

18 Mar 2021

PONE-D-20-31370R1

Treatment-seeking behaviour among people with opioid use disorder in the high-income countries: A systematic review and meta-analysis

PLOS ONE

Dear Dr. Hall,

Thank you for submitting your manuscript to PLOS ONE. After careful consideration, we feel that it has merit but does not fully meet PLOS ONE’s publication criteria as it currently stands. Therefore, we invite you to submit a revised version of the manuscript that addresses the points raised during the review process.

Reviewer #1 and #2 generally agree with the publication of the paper. However, the reviewer #3 still raises critical issues that needed to be further addressed. It is up to the authors whether or not to revise the paper again. I think that the Reviewer #3 has a point.

We look forward to receiving your revised manuscript.

Kind regards,

Kyoung-Sae Na, M.D.

Academic Editor

PLOS ONE

Reviewers' comments:

Reviewer's Responses to Questions

**Comments to the Author**

1. If the authors have adequately addressed your comments raised in a previous round of review and you feel that this manuscript is now acceptable for publication, you may indicate that here to bypass the “Comments to the Author” section, enter your conflict of interest statement in the “Confidential to Editor” section, and submit your "Accept" recommendation.

Reviewer #1: All comments have been addressed

Reviewer #2: (No Response)

Reviewer #3: (No Response)

2. Is the manuscript technically sound, and do the data support the conclusions?

Reviewer #1: Yes

Reviewer #2: Partly

Reviewer #3: No

3. Has the statistical analysis been performed appropriately and rigorously? 

Reviewer #1: Yes

Reviewer #2: I Don't Know

Reviewer #3: Yes

4. Have the authors made all data underlying the findings in their manuscript fully available?

Reviewer #1: No

Reviewer #2: Yes

Reviewer #3: Yes

5. Is the manuscript presented in an intelligible fashion and written in standard English?

Reviewer #1: Yes

Reviewer #2: Yes

Reviewer #3: Yes

6. Review Comments to the Author

Reviewer #1: The revised manuscript entitled “Treatment seeking behaviour among people with opioid use disorder in high income countries: A systematic review and meta-analysis” is significantly improved from the original submission. The authors have been comprehensive in their response to prior reviews. I have no additional major critiques. One suggestion would be to state in the abstract that while the search was intended to capture all high income countries only manuscripts from the US met inclusion criteria.

Reviewer #2: Upon re-reviewing this article I realized that the authors seem to use "treatment seeking" and "access to treatment" interchangeably. For instance, the Objective in the Abstract is to "To determine treatment seeking behaviour in those with opioid use disorder (OUD) in the high-income countries" but the authors present access to treatment. I do not think these 2 terms can be used interchangeably. It would be important for the authors to define/how they conceptualize "treatment seeking" and justify why they then move on to using the term "access". Alternatively, please remove the "access to treatment language".

Please also specify whether any relevant articles published before 2008 appeared in your search. If so, why were they excluded given that there were no year restrictions for your search.

Reviewer #3: The authors were very thoughtful in their responses to the reviewers. However, I still have significant concerns about the methodology that were not alleviated with inclusion of several items as limitations, but rather, call into question the appropriateness of the methodology. Some further detail and considerations may be warranted:

The authors note in their response that “only studies from the USA were available in the literature.” I believe there are still a number of issues that arise from this methodology and the objectiveness of this statement. Despite noting it in the limitations, the fact that only English language publications were used discounts a substantial amount of literature in high-income countries, and so continuing to use the title “High-income countries” is misleading to readers. In addition, there are nuances to other high income countries that may not be accounted for. For instance, were detoxification programs included? These are often referred to as simply ‘substance use detoxification’ programs. Would this be accounted for in the search criteria? Were studies on the prescribing of buprenorphine included? The Canadian Institute for Health publishes treatment-seeking statistics (i.e., proportion of Canadians starting opioid therapy) through assessments of prescriptions due to their ability to follow individuals through national health systems. It’s possible that the studies in the United States dominated the results because without nationalized healthcare, we rely on studies, as opposed to utilization of nationalized health systems for this information, which may see publication elsewhere than a study-based journal article. These are not just limitations, but raise questions about the appropriateness of this methodology. This could be why 11 of the 13 articles includes are from just two national studies in the US, which also raise questions about the generalizability of the information provided.

As noted in the previous review, NSDUH and NESARC are both interviews where an individual is asked to self-report behaviors, although the authors have updated the manuscript to note that they are asked specific criteria, here off the DSM. This leads to question of what the authors deemed to be “self-report” vs. “diagnosed” OUD that may have led to studies from other countries being excluded. Many other countries likely do not use the DSM, so there may be variations in how opioid use disorder is clinically diagnosed. In France, OUD is primarily treated by general practitioners who prescribe buprenorphine, and like Canada, this is tracked through the French National Health Insurance database. Did the authors only take articles where a diagnostic process was specifically outlined, or was a physician prescribing buprenorphine for OUD enough, on the assumption that the physician would not prescribe buprenorphine if there was not a clinical diagnosis? More information on needs to be provided on what constituted a “clinical diagnosis.”

7. PLOS authors have the option to publish the peer review history of their article (what does this mean?). If published, this will include your full peer review and any attached files.

Reviewer #1: No

Reviewer #2: No

Reviewer #3: No

---

## [Author Response · Author response to Decision Letter 1]

29 Apr 2021

Please see word document with response to reviewers comments

---

## [Decision Letter · Decision Letter 2]

13 May 2021

PONE-D-20-31370R2

Treatment-seeking behaviour among people with opioid use disorder in the high-income countries: A systematic review and meta-analysis

PLOS ONE

Dear Dr. Hall,

Thank you for submitting your manuscript to PLOS ONE. After careful consideration, we have decided that your manuscript does not meet our criteria for publication and must therefore be rejected.

I am sorry that we cannot be more positive on this occasion, but hope that you appreciate the reasons for this decision.

Yours sincerely,

Kyoung-Sae Na, M.D.

Academic Editor

PLOS ONE

Additional Editor Comments (if provided):

According to the reviewers, the revised version of the manuscript seems to be inappropriate for publication. Particularly, most of the methodological issues, which have been raised by the Reviewer 3, were not resolved.

Reviewers' comments:

Reviewer's Responses to Questions

**Comments to the Author**

1. If the authors have adequately addressed your comments raised in a previous round of review and you feel that this manuscript is now acceptable for publication, you may indicate that here to bypass the “Comments to the Author” section, enter your conflict of interest statement in the “Confidential to Editor” section, and submit your "Accept" recommendation.

Reviewer #2: (No Response)

Reviewer #3: (No Response)

2. Is the manuscript technically sound, and do the data support the conclusions?

Reviewer #2: Partly

Reviewer #3: Partly

3. Has the statistical analysis been performed appropriately and rigorously? 

Reviewer #2: I Don't Know

Reviewer #3: Yes

4. Have the authors made all data underlying the findings in their manuscript fully available?

Reviewer #2: Yes

Reviewer #3: Yes

5. Is the manuscript presented in an intelligible fashion and written in standard English?

Reviewer #2: No

Reviewer #3: Yes

6. Review Comments to the Author

Reviewer #2: I appreciate the authors taking the time to address my concerns. After this round of revision I must say these concerns have only been partially addressed. Namely, there continues to be confusion when discussing treatment seeking and unmet treatment needs. For instance, in the Conclusion of the Abstract the authors state that: "This review found that one in five people with OUD sought OUD treatment in the previous 12 months and two in five people with OUD sought OUD treatment in their lifetime. These results indicate a significant unmet treatment need for those with OUD, the reason for this low treatment seeking requires immediate attention and future work." To me, there is a logical gap about drawing conclusions on treatment gaps when the results presented pertain to treatment seeking. This argument would make sense if data presented pertained to people seeking treatment but not receiving it. We could then talk about an unmet need. I think it might be wise to revise the conclusion and focus on how we can move individuals toward treatment seeking. This misalignment is present throughout the manuscript.

Could you also please specify what is meant by treatment gap in the Discussion (line 305). Please also specify what other factors might be in line 330.

Reviewer #3: (No Response)

7. PLOS authors have the option to publish the peer review history of their article (what does this mean?). If published, this will include your full peer review and any attached files.

Reviewer #2: No

Reviewer #3: No

- - - - -

---

## [Author Response · Author response to Decision Letter 2]

5 Jul 2021

Please see rebuttal letter in the downloads section.

This has the responses to all issues raised by the reviewers.

---

## [Decision Letter · Decision Letter 3]

4 Oct 2021

Treatment-seeking behaviour among people with opioid use disorder in the high-income countries: A systematic review and meta-analysis

PONE-D-20-31370R3

Dear Dr. Hall,

We’re pleased to inform you that your manuscript has been judged scientifically suitable for publication and will be formally accepted for publication once it meets all outstanding technical requirements.

Kind regards,

Adam Todd, PhD

Academic Editor

PLOS ONE

Additional Editor Comments (optional):

I have reviewed this manuscript and I am satisfied that the authors responded to the comments in a robust and satisfactory way; the limitations are clearly described and the review has been reported according to the appropriate checklist.  In addition, the paper has been reviewed by in expert in statistical methods who is happy by the approach taken by the authors. Overall, I am therefore satisfied that this paper meets the threshold for publication in PLOS ONE.

Reviewers' comments:

Reviewer's Responses to Questions

**Comments to the Author**

1. If the authors have adequately addressed your comments raised in a previous round of review and you feel that this manuscript is now acceptable for publication, you may indicate that here to bypass the “Comments to the Author” section, enter your conflict of interest statement in the “Confidential to Editor” section, and submit your "Accept" recommendation.

Reviewer #4: All comments have been addressed

2. Is the manuscript technically sound, and do the data support the conclusions?

Reviewer #4: (No Response)

3. Has the statistical analysis been performed appropriately and rigorously? 

Reviewer #4: (No Response)

4. Have the authors made all data underlying the findings in their manuscript fully available?

Reviewer #4: (No Response)

5. Is the manuscript presented in an intelligible fashion and written in standard English?

Reviewer #4: (No Response)

6. Review Comments to the Author

Reviewer #4: (No Response)

7. PLOS authors have the option to publish the peer review history of their article (what does this mean?). If published, this will include your full peer review and any attached files.

Reviewer #4: No

---

## [Editor Report · Acceptance letter]

7 Oct 2021

PONE-D-20-31370R3 

Treatment-seeking behaviour among people with opioid use disorder in the high-income countries: A systematic review and meta-analysis 

Dear Dr. Hall:

I'm pleased to inform you that your manuscript has been deemed suitable for publication in PLOS ONE. Congratulations! Your manuscript is now with our production department. 

Kind regards, 

on behalf of

Dr. Adam Todd 

Academic Editor

PLOS ONE